# Radiometric Terrain Flattening of Geocoded Stacks of SAR Imagery

**Piyush S. Agram** *[ID], **Michael S. Warren** [ID], **Scott A. Arko** and **Matthew T. Calef** [ID]

Descartes Labs, Santa Fe, NM 87501, USA
* Correspondence: piyush@descarteslabs.com

**Abstract:** We have described an efficient approach to radiometrically flatten geocoded stacks of calibrated synthetic aperture radar (SAR) data for terrain-related effects. We have used simulation to demonstrate that, for the Sentinel-1 mission, one static radiometric terrain-flattening factor derived from actual SAR imaging metadata per imaging geometry is sufficient for flattening interferometrically compliant stacks of SAR data. We have quantified the loss of precision due to the application of static flattening factors, and show that these are well below the stated requirements of change-detection algorithms. Finally, we have discussed the implications of applying radiometric terrain flattening to geocoded SAR data instead of the traditional approach of flattening data provided in the original SAR image geometry. The proposed approach allows for efficient and consistent generation of five different Committee of Earth-Observation Satellites (CEOS) Analysis-Ready Dataset (ARD) families—Geocoded Single-Look Complex (GSLC), Interferometric Radar (InSAR), Normalized Radar Backscatter (NRB), Polarimetric Radar (POL) and Ocean Radar Backscatter (ORB) from SAR missions in a common framework.

**Keywords:** RTC; SAR; Sentinel-1

## 1. Introduction

Radiometrically terrain-flattened synthetic aperture radar (SAR) imagery is widely considered to be the first SAR-derived analysis-ready dataset (ARD) family [1] and is of considerable interest to the sustainability, ecosystem, and agriculture science communities. The Sentinel-1 SAR constellation [2], part of Europe's Copernicus Earth-Observation Programme, has significantly increased the adoption of SAR data within geospatial data frameworks, including data cubes, that were originally developed for optical imagery. Terrain-flattened SAR ARD datasets from other SAR missions such as ALOS-2 and NovaSAR are also under active development; see https://ceos.org/ard (accessed on 1 January 2023). A large number of national agencies and commercial entities now offer Sentinel-1 backscatter imagery as a foundational service on which to build applications, e.g., [3]. These ARD datasets are expected to support a wide number of applications, including land-cover classification, multi-temporal change detection, etc.

SAR systems collect imagery in side-looking geometries and the resulting radar backscatter is determined by both the imaging geometry and the material properties of the area being imaged. For example, hills facing the SAR system are typically brighter, as they scatter energy back toward the sensor, and hills facing away are darker as they scatter energy away from the sensor. Analytic applications are concerned with the material properties of the area being imaged, not the imaging geometry. Modeling and reducing the contribution of the imaging geometry in the observed backscatter signal is called terrain flattening. Most terrain-flattening processors implement the Gamma flattening approach [4] or related variants that have been optimized for performance, e.g., [5]. The Gamma flattening approach operates on Level-1 SAR imagery in slant-range or ground-range coordinate systems and generates geocoded, radiometrically terrain-flattened imagery. The rest of this manuscript assumes familiarity with the Gamma flattening formulation [4], as we have relied heavily on associated terminology and algorithm descriptions.

*Terminology*

Before proceeding, we describe some of the key terms used in this manuscript. We have attempted to stay consistent with [4] and with the terminology used by the Sentinel-1 toolbox [6] user community.

1. A geocoded product or *Geocoded Terrain-Corrected* (GTC) product, as referenced in the SAR mission product user guides, is derived by precisely geolocating SAR imagery using a Digital Elevation Model (DEM) [7]. Alternately, the GTC products themselves could be generated directly by focusing raw radar pulses onto a regular map grid [8]. When starting from Level-1 products, SAR imagery is usually calibrated to $\sigma_{0,E}$ before it is interpolated onto a regular map grid. Please note that the subscript $E$ in $\sigma_{0,E}$ indicates that the radiometry of the product has been adjusted under the assumption that the area being imaged lies on the reference ellipsoid or a well-defined flat reference surface. GTC products calibrated to $\beta_0$ or $\gamma_{0,E}$ are also common. GTC products have been geolocated precisely [9] but have not been corrected for terrain-related radiometric effects. Although we have provided mathematical expressions for working with different calibration levels of GTC products in this manuscript, we specifically focus on $\sigma_{0,E}$ GTC products, which we process at Descartes Labs at a global scale [10].

2. A terrain-flattened or *Radiometrically Terrain-Corrected* (RTC) product [4] or normalized radar backscatter (NRB) [11] product is a special type of GTC product where the imagery has been corrected for terrain-related radiometric effects. In the context of this manuscript, we always assume that an RTC product has been calibrated to $\gamma_{0,T}$ (Table I of [4]). RTC products are widely considered to be the most ready-for-analysis product derived from SAR imagery and most similar to optical imagery for developing similar applications [1,11]. The difference between GTC and RTC products is that the radiometry of GTC products corresponds to the reference ellipsoid or a reference flat surface and the radiometry of RTC products corresponds to the actual terrain represented by a DEM.

3. In general, a collection of GTC products generated on the same map grid is referred to as a *geocoded stack*. In the context of this manuscript, we specifically refer to GTC products generated on a common grid from interferometrically compliant acquisitions as a geocoded stack, unless mentioned otherwise. Such products are usually labeled with a common Path-Frame identifier (ERS, ALOS, etc.) or unique burst identifiers (Sentinel-1) [10,12]. These identifiers represent unique imaging geometry configurations, i.e., all images in the collection share baselines of less than a few kilometers with respect to each other and are acquired at similar incidence angles.

In this manuscript, we present a method to efficiently transform a GTC stack to an RTC stack using static flattening factors for SAR missions with orbit characteristics designed to consistently support interferometric analysis, such as Sentinel-1, ALOS-2, etc. We also present a testing framework, based on our proposed method using Sentinel-1 SAR metadata and apply it to ESA's Sentinel-1 toolbox. Finally, we discuss how our proposed method can significantly reduce computational resource requirements for generating global-scale Sentinel-1 RTC products while still generating these products in an inter-operable manner with other SAR-based ARD datasets.

## 2. Revisiting the Gamma Flattening Formulation

In this section, we have reinterpreted the Gamma flattening formulation [4] in the context of adopting the method to apply it directly to GTC products. We have tried to use the same terminology as presented in [4] to allow readers to relate to our interpretation more easily. We have ignored the issue of heteromorphism (layover) in this section, as this will be addressed in detail in Section 4.

### 2.1. Single DEM Facet

We have considered a single triangular facet of a DEM represented by points $T_{00}$, $T_{01}$ and $T_{10}$, with their projections onto the reference ellipsoid represented by points $E_{00}$, $E_{01}$ and $E_{10}$. Let $\theta_{inc}$ represent the local incidence angle of the facet at $T_c$, the centroid of $(T_{00}, T_{01}, T_{10})$. $T_{cc}$ is a point on the reference ellipsoid that projects to the same point in the slant-range geometry as $T_c$ and $\theta_0$ represents the nominal incidence angle at $T_{cc}$. $S_{00}$, $S_{01}$ and $S_{10}$ represents the projection of the DEM facet onto the slant-range plane, and $\psi_{prj}$ represents the projection angle [13], i.e., the angle between the normal to the DEM facet and the normal to the slant-range plane. Figure 1 depicts the geometry and is similar to a combination of Figures 2 and 5 of [4], except that the points $E_{00}$, $E_{01}$ and $E_{10}$ in Figure 5 of [4] represent regular grid points on a map for GTC products.

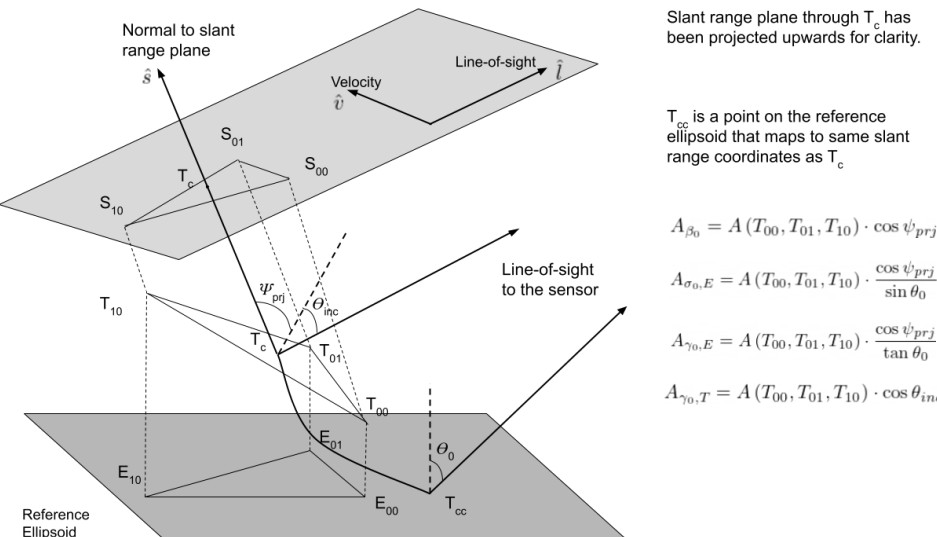

**Figure 1.** Normalization area relations for different SAR calibration levels of a single triangular DEM facet. The altitude of the DEM points $T_{00}$, $T_{01}$ and $T_{10}$ have been exaggerated for clarity. The normal vectors to the reference ellipsoid and the DEM facet are shown as dashed lines. The slant-range plane passes through $T_c$, but we have projected the plane upwards for clarity. $T_{cc}$ is a point on the reference ellipsoid that maps to the same slant-range coordinates as $T_c$, and the constant slant-range curve connecting these two points is shown. This image is comparable to Figures 2 and 5 of [4]. The unit vectors $\hat{l}$, $\hat{v}$ and $\hat{c}$ are used for defining interferometric baselines and are described in greater detail in Appendix A.

For the case where there exists a one-to-one mapping between pixels in the original radar image projection and a well-known map projection, the conservation of energy argument, as presented in [4], and the area relationships from Figure 1, show that

$$
\begin{aligned}
\gamma_{0,T} &= \beta_0 \cdot \frac{\left|\cos \psi_{prj}\right|}{\cos \theta_{inc}} \\
&= \sigma_{0,E} \cdot \frac{\left|\cos \psi_{prj}\right|}{\sin \theta_0 \cdot \cos \theta_{inc}} \\
&= \gamma_{0,E} \cdot \frac{\left|\cos \psi_{prj}\right|}{\tan \theta_0 \cdot \cos \theta_{inc}}
\end{aligned}
\tag{1}
$$

We have made the following observations about the formulation presented above:

1.  Equation (1) can be used to flatten GTC products corresponding to any of the standard calibration levels—$\beta_0$, $\sigma_{0,E}$ and $\gamma_{0,E}$.
2.  The formulation can be applied to GTC products in any well-known map projection system [14] as long as the actual area computations are performed in a 3-D

geocentric cartesian projection system, e.g., EPSG:4978, to avoid projection system related distortions.

3. Since the transformation of GTC products according to Equation (1) only involves computation of simple facet-by-facet area normalization factors (assuming no layover), we can significantly speed up processing and circumvent the use of large radar image index lookup tables.

4. If points $T_{00}$, $T_{01}$ and $T_{10}$ lie on the reference ellipsoid, $\psi_{prj}$ and $\theta_{inc}$ are complementary (see Equation (15) of [5]) and Equation (1) reduces to the classic Equations (2) and (3) of [4], i.e.,

$$\gamma_{0,E} = \beta_0 \cdot \tan \theta_0 = \frac{\sigma_{0,E}}{\cos \theta_0} \tag{2}$$

The calculation of area normalization factors in Equation (1) can be easily implemented within any open SAR processing software [6,15,16] that includes functions for interpreting SAR imaging geometry, in combination with a map projection transformation library such as PROJ [14].

*2.2. Extension to Rectangular Pixels*

The traditional Gamma flattening approach [4,5] (see Figures 1 and 2 of [5]) tracks the mapping between individual DEM facets and pixels in slant-range radar geometry using large lookup tables to account for the many-to-one mapping of area in map coordinates to slant-range geometry. The total area of all contributing facets is accumulated before imagery in $\beta_0$ is normalized to $\gamma_{0,T}$. This many-to-one mapping between map coordinates and slant-range coordinates almost always exists for each DEM facet as these two coordinate systems are not aligned. The use of large lookup tables significantly increases memory requirements and hinders parallelization of the flattening approach by requiring multiple passes over the map and radar image grids. When we start with GTC products, the calibrated backscatter measurements have already been aligned in map coordinates which presents an opportunity to significantly simplify the bookkeeping.

Extending Equation (1) for use with a rectangular GTC pixel is straightforward using area summation of facets visible to the radar (Equation (11) of [4]). The DEM is oversampled in the same projection system as the GTC product, such that an integer number of DEM pixels fits along each axis of the rectangular GTC pixel. The same caveats related to DEM oversampling as described in [4,5] apply to our formulation as well.

Interpreting our approach in the context of the mapping between slant-range coordinates and map coordinates used in the traditional Gamma flattening approaches [4,5], we observed that the integer relationship between the oversampled DEM pixel size and GTC pixel size guarantees that each DEM facet maps completely within the bounds of a single GTC pixel and that the sum of the projected area of all facets corresponding to a GTC pixel represented its total projected area, thus eliminating the need for using large lookup tables to track the contribution of the facet across multiple GTC pixels. This also allows trivially parallel implementations of our approach at a GTC pixel level. The facet area summation approach can be formulated as:

$$
\begin{aligned}
\gamma_{0,T} &= \beta_0 \cdot \frac{\sum_{\mathfrak{F}_v} A(T_{00}, T_{01}, T_{10}) \cdot |\cos \psi_{prj}|}{\sum_{\mathfrak{F}_v} A(T_{00}, T_{01}, T_{10}) \cdot \cos \theta_{inc}} \\
&= \frac{\sigma_{0,E}}{\sin \theta_0} \cdot \frac{\sum_{\mathfrak{F}_v} A(T_{00}, T_{01}, T_{10}) \cdot |\cos \psi_{prj}|}{\sum_{\mathfrak{F}_v} A(T_{00}, T_{01}, T_{10}) \cdot \cos \theta_{inc}} \\
&= \frac{\gamma_{0,E}}{\tan \theta_0} \cdot \frac{\sum_{\mathfrak{F}_v} A(T_{00}, T_{01}, T_{10}) \cdot |\cos \psi_{prj}|}{\sum_{\mathfrak{F}_v} A(T_{00}, T_{01}, T_{10}) \cdot \cos \theta_{inc}},
\end{aligned}
\tag{3}
$$

where $\mathfrak{F}$ represents all the facets of oversampled DEM pixels associated with the GTC pixel of interest and $\mathfrak{F}_v$ represents a subset of $\mathfrak{F}$ that is visible to the imaging SAR sensor. A facet is considered visible to the SAR sensor if it is not in active radar shadow and $\theta_{inc} < \theta_{thr}$.

$\theta_{thr}$ is usually set to $\cos^{-1}(0.05)$, consistent with the area ratio threshold in Section II-H of [4] or more conservatively to 85°, e.g., in [17]. Please note that this threshold is to avoid amplification of the noise due to the $\cos\theta_{inc}$ term in the denominator. We do not need similar thresholds for the $\cos\psi_{prj}$ term in the numerator but we do use its absolute value as projection angles can be greater than 90° when the facet itself is impacted by foreshortening. In general, $\theta_0$ and the look vector to the center of the pixel could be reused with all the contributing facets to further reduce the number of computations involved. [5,18] described a similar formulation but used a constant $\psi_{prj}$ and $\theta_{inc}$ for the entire rectangular pixel in slant-range coordinates. The formulation presented here operates on geocoded pixels and accounts for contribution from individual DEM facets.

We discuss a couple of practical examples of our DEM oversampling scheme here. A source DEM of 30 m posting is assumed for both examples. For terrain flattening a 10 m Sentinel-1 GTC product on a UTM grid, we recommend oversampling the source DEM to a spacing of 5 m at least on the same UTM grid and using a two-facet approach with each 5 m DEM pixel resulting in a total of eight facets in a 10 m GTC pixel. For terrain flattening a 100 m GTC product on a UTM grid, we recommend oversampling the source DEM to a spacing of 25 m at least on the same UTM grid and using a two-facet approach with each 25 m DEM pixel resulting in a total of 32 facets in a 100 m GTC pixel.

## 3. Terrain Flattening of Geocoded Stacks

In Section 2, we described an approach to terrain-flatten a single pixel of a GTC product. In this section, we have analyzed the sensitivity of the area-flattening factor from Equation (1) to the variation of imaging geometry within a stack for actual SAR acquisitions from sensors such as Sentinel-1 and ALOS-1 (see Figure 2). In this section, we again considered a single triangular DEM facet and study the effect of variations in $\theta_0$, $\theta_{inc}$, and $\psi_{prj}$. The backscatter term in Equation (1) is obtained from the source GTC product that we want to transform and the area term $A(T_{00}, T_{01}, T_{10})$ is independent of the imaging geometry. For all the examples presented in this section, we considered a facet located at near-range of the imaged swath as the impact of the change in imaging geometry decreases with slant range [19]. We also specifically considered the transformation of $\sigma_{0,E}$ GTC products to $\gamma_{0,T}$ as this is most relevant for use with our global-scale radar backscatter product [10]. This analysis can be easily extended in the same framework to study the transformation of $\beta_0$ and $\gamma_{0,E}$ GTC products.

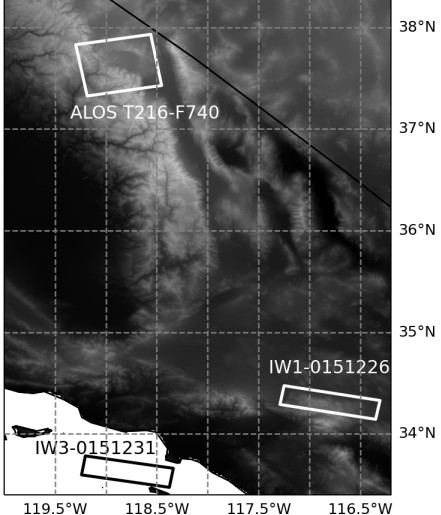

**Figure 2.** Footprints of the three stacks of SAR metadata that were used in this manuscript overlaid on the 1 km GLOBE DEM—Sentinel-1 burst IW1-0151226 over Big Bear in California, Sentinel-1 burst IW3-015131 over the ocean and ALOS Track 216, Frame 740 over Long Valley in California.

### 3.1. Sentinel-1

We considered a stack of 58 acquisitions corresponding to a single Sentinel-1 burst footprint with European Space Agency (ESA) identifier IW1-0151226 (Descartes Labs identifier [10] 071-2637-IW1-VV-RD) over Big Bear, California (see Figure 2) in the USA and spanning the time period of 1 January 2020 to 1 January 2021. The perpendicular baseline (see Appendix A for definition) variation of this stack was about 200 m, consistent with 100 m orbital tube radius RMS value for Sentinel-1 [2]. Note here that the choice of this footprint was not critical since the experimental setup does not involve actual imagery or an actual DEM. We have provided detailed burst information to let readers replicate the experiment, if they desire. This footprint is imaged from a right-looking descending geometry and we set up a single DEM facet as follows:

$$
\begin{aligned}
T_{00} &= (X_0, Y_0, 0) \\
T_{01} &= (X_0, Y_0 + 100, h_1) \\
T_{10} &= (X_0 - 50, Y_0 + 50, h_2),
\end{aligned}
$$

where $X_0$ and $Y_0$ represent the Easting and Northing coordinates of a nominal near-range pixel of the GTC product in a Universal Transverse Mercator (UTM) system. We varied $h_1$ in the interval $[-500, 500]$ meters and $h_2$ in the interval $[-250, 250]$ meters, to study the impact of imaging geometry in the relationship between $\gamma_{0,T}$ and $\sigma_{0,E}$. We note that the altitude of point $T_{00}$ is fixed to zero for the simulations presented here and moving the entire facet up or down by a constant within the limits of earth's topography ($-500$ to 9000 m) only modifies the mean incidence angles for spaceborne missions, and does not affect the interpretation of our results. We also note that the spacing in Easting and Northing of the vertices are not critical as long as they are comparable to that of typical DEMs or GTCs. Area normalization factors are sensitive to the slope of the DEM facet and one can regenerate the same results by scaling up the range of $h_1$ and $h_2$ in accordance with the desired spacing. This current setup represents one of the four facets including the center point of a 100 m GTC pixel following [5].

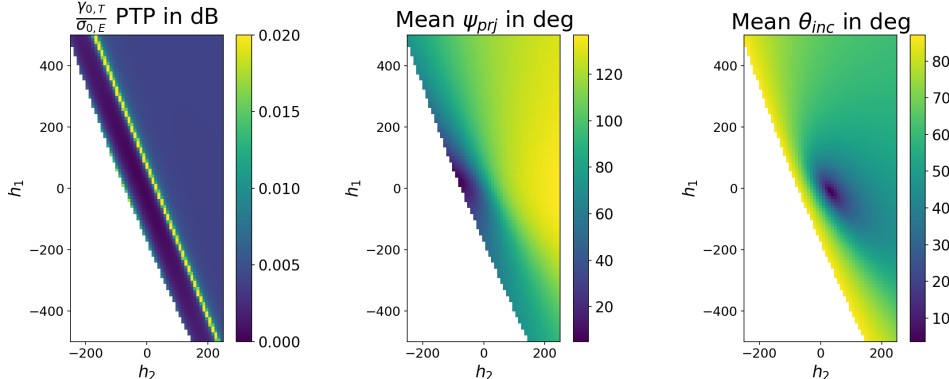

**Figure 3.** (**Left**) Peak-to-peak variation in $\gamma_{0,T}/\sigma_{0,E}$ in dB, (**Middle**) Mean projection angle ($\psi_{prj}$) and (**Right**) Mean local incidence angle ($\theta_{inc}$) as a function of $h_1$ and $h_2$ in meters of a single facet corresponding to a stack of 58 Sentinel-1 acquisitions corresponding to burst footprint IW1-0151226. The bright line in (**Left**) corresponds to $\psi_{prj} = 90°$. Plots have been masked for the region corresponding to $\cos\theta_{inc} > \cos^{-1}(0.05)$.

Figure 3 shows that for this stack, for most local incidence angles the variation of $\gamma_{0,T}/\sigma_{0,E}$ was well below 0.01 dB and we only started approaching 0.02 dB variation for local incidence angles greater than 85 degrees. The bright strip in Figure 3 (Left) corresponds to $\psi_{prj} = 90°$ and is a numerical artifact from estimating a ratio of two small numbers. When interpreted in the context of Equation (3), facets with $\psi_{prj} = 90°$ would contribute very little to the sum in the numerator. The perpendicular baseline spread in this stack

is comparable to other Sentinel-1 stacks due to the mission's narrow orbital tube [2]. The observed variation was much smaller than the typical relative radiometric accuracy of 0.1 dB for Sentinel-1 [20], even for fairly steep terrain, i.e., over a wide range of $h_1$ and $h_2$. Consequently, we can conclude that a constant pixel-by-pixel flattening factor (per imaging geometry) is sufficient to transform $\sigma_{0,E}$ GTC products to $\gamma_{0,T}$ for Sentinel-1 stacks. We extend this argument to suggest that a constant pixel-by-pixel flattening factor (per imaging geometry) is sufficient to efficiently transform GTC products to RTC products from other SAR missions with narrow orbital tubes such as ALOS-2 and NISAR as well.

Another interpretation of the result above is that the area projections of DEM facets do not vary significantly between passes for missions with narrow orbit tubes. However, the fact that each Level–1 SAR product is distributed in its own projection system [10] requires users to expend computational resources and build elaborate lookup tables, from these projection systems to well-known map coordinate systems, to carefully account for area contribution to each SAR product pixel. Our proposed approach of starting from a GTC product eliminates the need for mapping area projections for each product. A static terrain-flattening factor layer can be built using a reference orbit and this correction factor can be reused for all GTC SAR products acquired from a similar imaging geometry. Additionally, this framework allows us to estimate errors introduced using static terrain-flattening factors using actual orbit data.

Navacchi et al. [19] also observed that the variation in projected incidence angle is on the order of 0.005 degrees (standard deviation) for Sentinel-1, which is consistent with our simulations, and proposed the use of static correction factors to transform $\sigma_{0,E}$ to projected local incidence angle (PLIA) normalized backscatter product. With our simulations, we can show that the same approach can be used for generating $\gamma_{0,T}$ RTC products on the fly as well.

### 3.2. ALOS-1

Although we considered a narrow orbital tube mission in Section 3.1, here we considered a stack of 34 ALOS-1 acquisitions corresponding to Path 216, Frame 740 over Long Valley, California, USA (see Figure 2) and spanning the time period of 1 June 2006 to 1 March 2011 and exhibiting a perpendicular baseline variation about 6500 m (see Figure 4). We chose this specific track, which covers the Rosamond corner reflector site, as it was acquired on every ascending pass by the ALOS-1 mission for calibration and validation activities. This frame is imaged from a right-looking ascending geometry and we set up a single DEM facet as follows:

$$
\begin{aligned}
T_{00} &= (X_0, Y_0, 0) \\
T_{01} &= (X_0, Y_0 + 100, h_1) \\
T_{10} &= (X_0 + 50, Y_0 + 50, h_2)
\end{aligned}
$$

where $X_0$ and $Y_0$ represent the Easting and Northing coordinates of a nominal near-range pixel of the GTC product in a Universal Transverse Mercator (UTM) system. Note the change in Easting of $T_{10}$ to accommodate the change in the pass direction.

Figure 5 shows that for this stack, the variation in area normalization factor in Equation (1) was as high as 0.3 dB which is larger than the typical radiometric requirement of 0.1 dB. Even in this case, for most incidence angles the observed variation was below 0.15 dB but we saw significantly higher variation for incidence angles greater than 80 degrees. The band corresponding to $\psi_{prj} = 90°$ is also clearly visible and is broader than the Sentinel-1 case due to larger variation in imaging geometry. In general, we can deduce that a constant pixel-by-pixel correction factor per imaging geometry is insufficient to transform $\sigma_{0,E}$ GTC products to $\gamma_{0,T}$ for ALOS-1 stacks or missions with wider orbital tubes in general. In Section 3.3 we present a solution for these types of missions.

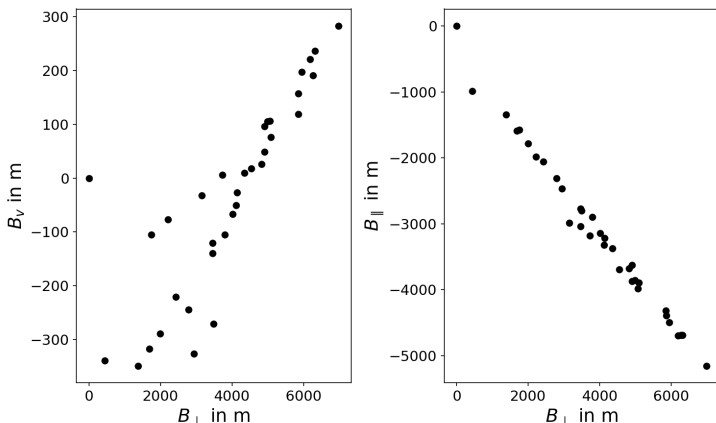

**Figure 4.** Scatter plots of (**Left**) along-track baseline ($B_v^{ref}$) in meters vs. perpendicular baseline ($B_\perp^{ref}$) in meters and (**Right**) parallel baseline ($B_\parallel^{ref}$) in meters vs. perpendicular baseline ($B_\perp^{ref}$) in meters corresponding to ALOS-1 stack w.r.t the imaging geometry of 2006-06-30 acquisition for $h_1 = 0$ and $h_2 = 0$.

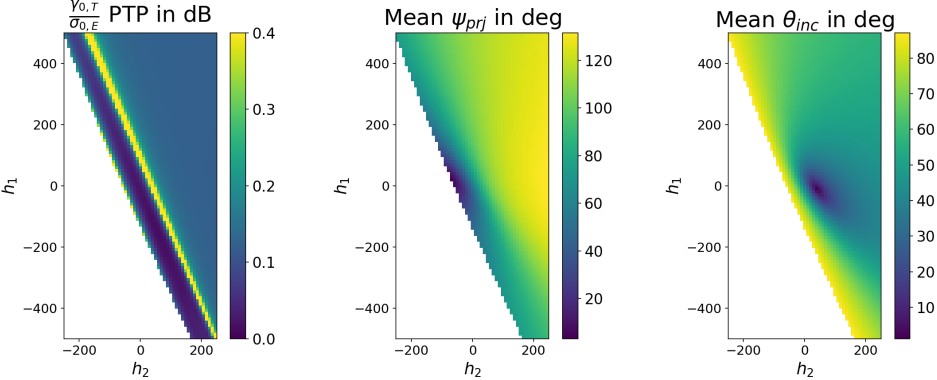

**Figure 5.** (**Left**) Peak-to-peak variation in $A_{\gamma_{0,T}}$, (**Middle**) Mean projection angle ($\psi_{prj}$) and (**Right**) Mean local incidence angle ($\theta_{inc}$) as a function of $h_1$ and $h_2$ in meters of a single facet corresponding to a stack of 34 ALOS-1 acquisitions corresponding to Path 216, Frame 740. The bright line in (**Left**) corresponds to $\psi_{prj} = 90°$. Plots have been masked for the region corresponding to $\cos\theta_{inc} > \cos^{-1}(0.05)$.

*3.3. Generalized Formulation*

In this section, we have presented a generalized formulation, inspired by SAR interferometry (InSAR), that allows us to exploit interferometric baseline information (see Appendix A) with static terrain-flattening terms to efficiently flatten GTC products from InSAR-capable SAR missions. Since, this manuscript focuses on SAR missions with narrow orbit tubes such as Sentinel-1, ALOS-2, and NISAR, we only present the motivation for the formulation without delving into greater detail and analysis.

Although we showed in Section 3.2 that a constant pixel-by-pixel flattening factor is insufficient for flattening stacks with large baseline variations, we also observe that the form of Equation (1) suggests that there might be a relationship between interferometric baselines (see Appendix A) and $\gamma_{0,T}/\sigma_{0,E}$, akin to the topography phase term used in differential InSAR analysis, e.g., [21], due to the dependence on the incidence angle term $\theta_{inc}$ and the projection angle term $\psi_{prj}$. Following a similar Taylor-series expansion as Equations (8)–(10) in [21] we can generalize that for each GTC pixel

$$\frac{\gamma_{0,T}}{\sigma_{0,E}}(\text{in dB}) \approx \frac{\gamma_{0,T}^{ref}}{\sigma_{0,E}^{ref}}(\text{in dB}) + \mathcal{C} \cdot B_\perp^{ref} + \mathcal{D} \cdot B_{\mathfrak{v}}^{ref} \tag{4}$$

where $\gamma_{0,T}^{ref}/\sigma_{0,E}^{ref}$ represents the ratio computed using a reference imaging geometry (could be a reference orbit or a reference scene in the same stack) in decibels, $B_{\perp}^{ref}$ and $B_{\mathfrak{v}}^{ref}$ represent the perpendicular and along-track baselines of the GTC pixel in a product of interest regarding the reference imaging geometry, and $\mathcal{C}$ and $\mathcal{D}$ represent a constant scaling factor associated with the GTC pixel. The parallel baseline ($B_{\parallel}^{ref}$) does not contribute to Equation (4) as this term does not modify the line-of-sight vector and its impact on $\theta_0$ is minimal. The scaling factors, $\mathcal{C}$ and $\mathcal{D}$, for each GTC pixel depend on the slope of facets associated with the pixel and can be easily computed numerically, contemporaneously with the computation of $\gamma_{0,T}^{ref}/\sigma_{0,E}^{ref}$, by recomputing $\theta_{inc}$ and $\psi_{prj}$ after perturbing the estimated reference satellite location by unit perpendicular and along-track baseline vectors. For modern sensors with Doppler steering capability, e.g., Sentinel-1, ALOS-2, TERRASAR-X etc., the along-track baseline is in the order of a few meters and the associated term ($\mathcal{D}$) can be ignored in Equation (4).

We have demonstrated the linear relationship between $\gamma_{0,T}/\sigma_{0,E}$ and perpendicular baseline ($B_{\perp}$) in Figure 6 for 3-different sets of combinations of $h_1$ and $h_2$ in the ALOS-1 simulation from Section 3.2. The minor deviations from the linear trend that we observed were due to the orientation of the facets and the resulting sensitivity to the along-track baseline, which can be as high as 600 m for ALOS-1 (Figure 4 (Left)). Although we demonstrated this relationship with a single triangular DEM facet, the idea can be extended to area summation of facets in Equation (3). In general, we can deduce that in addition to a constant flattening factor computed using a reference imaging geometry, similar to Section 3.1, an additional pixel-by-pixel perpendicular baseline-related scale factor ($\mathcal{C}$ in Equation (4)) should be sufficient to transform $\sigma_{0,E}$ GTC products to $\gamma_{0,T}$ for ALOS-1 stacks. Using the perpendicular baseline-related scale factor with Sentinel-1 (Section 3.1) or ALOS-1 (Section 3.2) stacks increases the precision of radiometric correction to well below 0.005 dB, even for incidence angles greater than 85 degrees in our simulations. Similar baseline terms can also be used to extend the approach proposed in [19] for use with other SAR missions. We also note that if the orbital tube for Sentinel-1 mission were to be relaxed at a future date due to operational constraints, the presented formulation can easily accommodate it and efficient terrain flattening can still be achieved by incorporating the baseline dependent terms. The rest of the manuscript will focus on using static terrain-flattening factors with the Sentinel-1 mission.

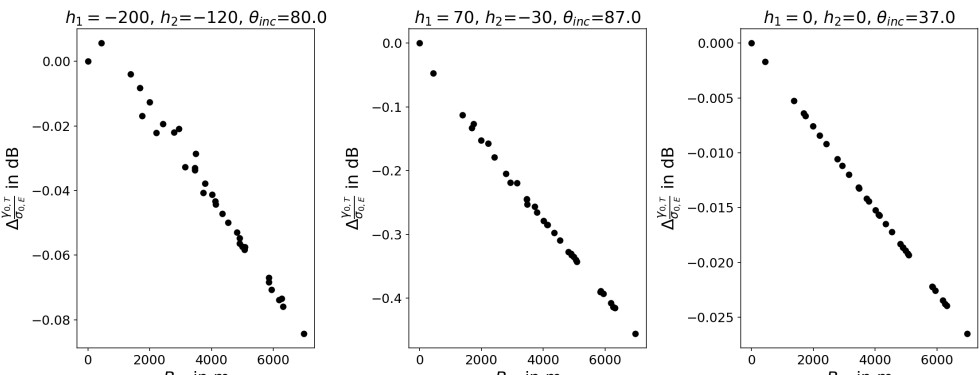

**Figure 6.** Scatter plot of $\Delta A_{\gamma_{0,T}}$ in dB vs. perpendicular baseline ($B_{\perp}$) corresponding to ALOS stack used in Section 3.2 for three different facets. The values of $h_1$ and $h_2$ for the facet, along with the resulting local incidence angle $\theta_{inc}$ are also shown.

## 4. Impact of Layover

Radar shadow and layover are two inherent limitations of side-looking SAR imaging systems, e.g., [22]. Regions impacted by radar shadow have to be masked out from SAR imagery as no energy is scattered back to the SAR sensor from these areas and the values for these pixels in GTC products are essentially noise. As far as we can tell,

there is no disagreement among the user community regarding the masking approach to radar shadow.

The original Gamma flattening approach [4] proposes a solution, for regions impacted by layover or heteromorphism. We have attempted to lay out the inherent assumptions of this approach and describe the impact of correcting for heteromorphic effects on an image-by-image basis on a stack of geocoded imagery on multi-temporal change-detection applications. Specifically, we have made the case for the complete masking of pixels impacted by layover compared to using the computationally expensive approach of tracking all facets in each SAR image.

### 4.1. Single SAR Image

Layover is a bigger challenge with airborne SAR data rather than spaceborne SAR data, which is acquired with nominal incidence angles of 23 to 45 degrees (in most modern spaceborne SAR sensors [23]), due to the comparatively low altitude of the airborne sensing platform. For Sentinel-1, the fraction of data affected by layover for a region such as the United Kingdom is below one percent [24] and could be as high as ten percent when specifically looking at targeted areas of interest (AOI) with steep topography [25]. The fraction of pixels that are impacted by layover or shadow in all corresponding Sentinel-1 imaging geometries is very low, e.g., [26,27].

The Gamma flattening approach [4,5] rigorously tracks the topological relationship between map coordinates and slant-range radar geometry to account for the many-to-one mapping of energy in map coordinates to slant-range geometry. The total area of all contributing facets is accumulated before imagery in $\beta_0$ is normalized to $\gamma_{0,T}$. However, the starting point of our proposed approach is a GTC product and not an image in slant-range geometry implying that we suffer from the issue indicated in Section II-D of [4]. Pixels impacted by layover stand out as too bright in GTC products as observed in a number of previous works, e.g., [4,28].

The conventional terrain-flattening approach corrects for the layover effect under a very specific assumption of a distributed scattering mechanism. It is assumed that the scattering mechanism in all pixels on the map contributing to the same slant-range pixel is the same and hence their relative contributions are proportional to the projected area. This assumption is not necessarily true in most real-life scenarios. For example, consider the textbook case of layover in SAR imagery where the top of a hill and its base are at the same slant range from the imaging platform. The conventional terrain-flattening approach redistributes energy under the assumption that the material and scattering mechanisms are the same at all the geographic locations mapping to the same pixel in slant-range geometry. Although this approach reduces the number of bright outliers and improves the histogram, as shown in [4], it may not be correct.

### 4.2. Stack of SAR Images

Stacks of SAR imagery are often used in the context of multi-temporal change detection, e.g., [29]. In such scenarios, careful handling of regions affected by layover, which cannot be corrected, become important to avoid false detections and to improve the robustness of any analysis building on SAR backscatter imagery [3,28]. In Section 4.1, we highlighted that energy is redistributed proportional to the area contributions for layover regions in the Gamma flattening approach. Although this approach improves the histogram and visualization of a single image, it does not guarantee consistent redistribution of energy over time in a stack of RTC products. Our simulations in Section 3 show that if the same facets contributed to a layover-affected pixel in every image in the stack, their relative area contributions will be consistent. However, slight variations in satellite position between passes results in different facets contributing to layover-affected regions at different time epochs, resulting in an inconsistent redistribution of energy over time. In fact, SAR tomography techniques have the potential to coherently redistribute energy in layover regions by exploiting the variation in imaging geometry in a stack of SAR images,

but this topic is beyond the scope of this manuscript. The need for masking out layover regions has also been empirically observed by several research groups, across a spectrum of applications—e.g., [30,31].

For regions not impacted by layover, the assumption of distributed scattering mechanism could still be physically wrong, particularly in heterogeneous terrain as in urban areas. However, our simulations in Section 3 show that terrain flattening can be interpreted as a pixel-by-pixel scaling within a stack, and relative changes in time are preserved for such regions.

### 4.3. Shadow–Layover Mask

Several shadow–layover mask generation methodologies have been developed, in both slant-range geometry [22] as well as in map coordinates directly [28]. It is worth noting that the shadow–layover mask varies slightly between acquisitions in a Sentinel-1 stack due to variations in platform imaging positions. Having collectively analyzed several stacks of shadow–layover masks spanning the years 2021 and 2022 and corresponding to the same imaging geometry for Sentinel-1, in our experience, it suffices to use the shadow–layover mask of a single reference acquisition and buffer it by 150–200 m, comparable to Sentinel-1 mission's baseline variation, before applying it to the whole stack during change-detection analysis. This buffering also accounts for imperfections in the source shadow–layover masks which are typically generated using a simple ray tracing method. Our approach to buffering the shadow–layover mask is similar to [28], but is meant to compensate for variations of imaging geometry within the orbit tube in the stack of Sentinel-1 images, and the original mask is derived in radar coordinates using the traditional approach. In this work, we do not delve deeper into the accuracy of different approaches of generating a shadow–layover mask but emphasize that we can significantly reduce the amount of computation needed by exploiting the narrow orbital tube of Sentinel-1 and using a buffered shadow–layover mask from a reference acquisition or orbit for the whole stack. In the context of our proposed flattening approach in Section 2 and Equation (3), a GTC pixel should be considered to be impacted by shadow or layover even if a single contributing facet is affected.

## 5. Experiments with the Sentinel-1 Toolbox

In Sections 2 and 3, we presented simulations using individual DEM facets, and the results justify the use of static factors for terrain-flattening Sentinel-1 imagery. In this section, we have described experiments that we conducted with the Sentinel-1 toolbox [6] to validate our simulations. We emphasize that we chose the Sentinel-1 toolbox because of its accessibility and the fact that it is widely used, allowing any interested reader to replicate these experiments. We believe that results from similar experiments with other open or commercial software will be of interest to the end-user community, especially if a large volume of RTC data are going to be generated systematically with the software.

We have used the following experimental setup with the Sentinel-1 toolbox (SNAP version 9.0.0). We staged Sentinel-1 Single-Look Complex (SLC) products corresponding to the same imaging geometry (burst footprints) and replaced the imagery arrays in the TIFF files with a constant Digital Number (DN) value of 8000. We also staged the same global DEM that we use for processing our global backscatter and InSAR products [10] and used it as an input to Sentinel-1 toolbox workflows. We used standard workflows [32] to generate geocoded layers for calibrated $\sigma_{0,E}$, terrain-flattened $\gamma_{0,T}$ and shadow–layover masks for each of these products. The choice of this constant value is not critical, as it cancels out when computing the ratio between $\gamma_{0,T}$ and $\sigma_{0,E}$. The geocoded products were generated on the same 10 m regular grid in a Universal Transverse Mercator (UTM) coordinate system as Sentinel-2, which we also use for our global-scale backscatter products [10]. No thermal noise correction was applied, as these experiments are designed to purely capture the impact of variation of imaging geometry within a stack. We then analyzed the statistics of the ratio of $\gamma_{0,T}$ to $\sigma_{0,E}$ using the geocoded products across the footprint and over time.

We repeated the experiment with different values of the *oversamplingMultiple* parameter, ranging from one to four in the terrain-flattening module to understand its impact on the processing errors, following the observations of [5]. We expect to see variations in this ratio across pixels in line with our simulations in Section 3.1, but we expect this ratio to be nearly constant in time.

As a constant DN was used instead of real SAR imagery, the resulting ratios should represent the consistency between SNAP and the method proposed here in the computation of projected facet areas and their summation. The other benefits of using constant DN imagery for this experiment include:

1.  The elimination of the effect of differences introduced by InSAR-grade interpolators [33] used in the complex-value interpolation of SLC data and noisier bilinear or bicubic interpolators used with real-valued intensity data in the workflow, letting us focus on geometric inconsistencies.
2.  A GTC $\beta_0$ product derived from a constant DN image in slant-range coordinates, which will also be a constant-valued image, thus allowing us to compare outputs with terrain-flattened products generated from GTC products as described in Section 2.
3.  The elimination of the effects introduced by inconsistent spatial averaging due to the use of a multilooking operator in slant-range coordinates, as multilooked products of constant DN images are also constantly valued. This effect is similar to phase-closure artifacts observed in pair-by-pair InSAR analysis as described in [10].

These different effects also contribute in their own way to the overall processing-error budget, but quantifying them is beyond the scope of this manuscript.

### 5.1. Open Ocean

The first example is a test over the open ocean off the coast of California, USA (see Figure 2)—ESA identifier IW3-0151231 (Descartes Labs identifier 071-2655-IW3-VV-RD)—with a stack of 38 acquisitions spanning the time period of 1 January 2020 to 1 January 2021. We picked an open-ocean region as this would not be impacted by shadow–layover effects or the quality of DEM being used. We would expect our results to mimic our simulations in Section 3.1. We observed a variation of $\sim$20 m in the reference terrain height values in the individual burst metadata, which introduces a small variation in calibrated $\sigma_{0,E}$ values. This variation can be interpreted as an additional small baseline variation term and from Section 3.1; we know that the effect of such a small baseline variation is negligible.

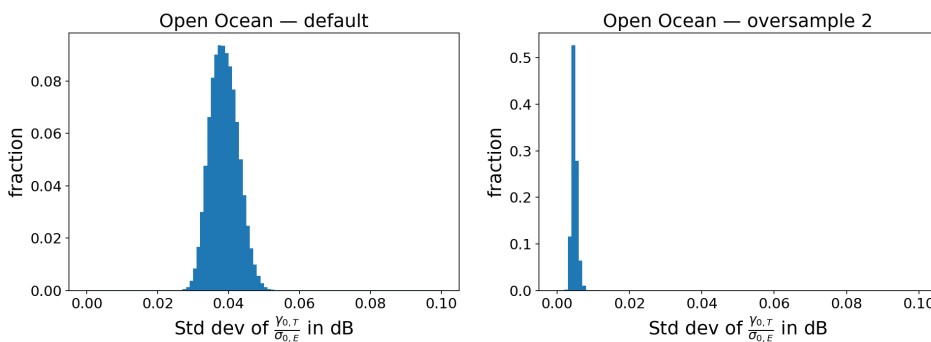

**Figure 7.** Histogram of the standard deviation of $\gamma_{0,T}/\sigma_{0,E}$ for all pixels in the open-ocean burst off the California coast with default oversampling factor (**Left**) and an oversampling factor multiple of 2 (**Right**). The results from an oversampling factor of 2 match our simulations from Section 3.1.

Figure 7 (Left) shows a clean peak around 0.04 dB for the histogram of pixel-by-pixel standard deviation of the ratio between $\gamma_{0,T}$ and $\sigma_{0,E}$. This number is larger than the 0.01 dB (peak-to-peak) variation from the simulations in Section 3.1. However, Sentinel-1 toolbox performance significantly improved when we increased the DEM oversampling by a multiple of 2, as in Figure 7 (Right). The peak was observed to be around $\sim$0.005 dB and is

well within acceptable limits for change-detection applications. This experiment confirmed the fact that using static flattening factors should be more than sufficient (<0.1 dB) for regions not impacted by shadow–layover. This type of metric would be useful for the end-user community to understand change-detection sensitivity of operational processing workflows that will be deployed for generating various CARD4L normalized radar backscatter or ocean radar backscatter products [11].

### 5.2. Rugged Terrain

We repeated the same experiment with our original burst footprint from Section 3.1 over Big Bear, California. This region is characterized by steep terrain as well as some flat regions. The Sentinel-1 toolbox estimated that only 1% of this footprint would be impacted by layover, and that a negligible area is impacted by shadow. We masked out the estimated ratios with a shadow–layover mask, using a buffer of 150 m as described in Section 4, before generating the multi-temporal statistics.

Based on our simulations in Section 3.1, we expected the histogram from this region to look similar to that of the open-ocean region above but with a slightly broader distribution as there are additional DEM interpolation operations involved which could introduce numerical noise. Although we observed a nice sharp peak around 0.04 dB just as for open ocean, we also observed a long tail extending to about 0.3 dB in Figure 8 (Left) with default settings. The standard deviation of the ratio for the majority of the image (83%) was below 0.1 dB. At a DEM oversampling multiple of 4, Figure 8 (Right), we observed that the peak was centered around $\sim$0.025 dB and the standard deviation of 87% of the image was below 0.1 dB. However, the tail of the distribution still extended to 0.3 dB.

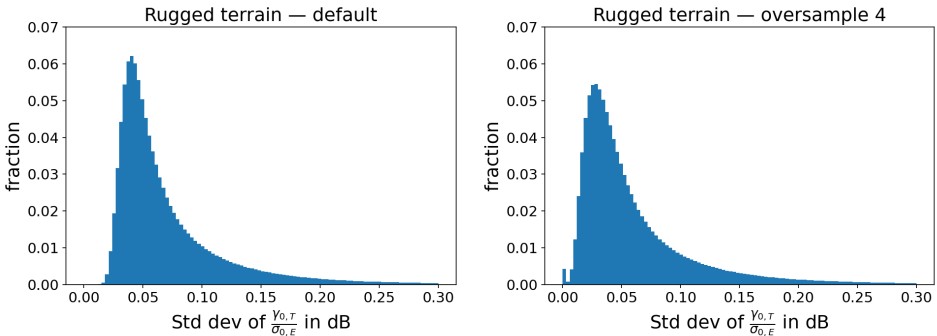

**Figure 8.** Histogram of the standard deviation of $\gamma_{0,T}/\sigma_{0,E}$ for all pixels in a rugged terrain burst over Big Bear, California with default oversampling factor (**Left**) and an oversampling factor multiple of 4 (**Right**). Data were masked with the shadow–layover mask before the histograms were estimated, but the long tail of the histogram for the rugged terrain burst indicates that there could be other subtle processing effects to account for in steep terrain.

The pixels contributing to the long tail of the histograms in Figure 8 were correlated with rugged terrain. We include Figure 9 over a particularly rugged region of the burst footprint to illustrate the correlation between observed standard deviation and the DEM, which indicates that this was likely a systematic processing artifact in Sentinel-1 toolbox. Another possible reason for the observed correlation is the underestimation of the areas impacted by shadow–layover. A number of additional software implementation factors can contribute to this observation, including thresholds chosen for the convergence of geometry computations, the possible use of single precision representations for certain intermediate computations, the handling of map projections, etc.

It is clear from comparing the histogram and associated image of the standard deviation of $\gamma_{0,T}/\sigma_{0,E}$, i.e., the histogram in Figure 8 (Left) with the image in Figure 9 (Left) and the histogram in Figure 8 (Right) with the image in Figure 9 (Middle), that oversampling of the DEM is critical for reliable terrain flattening, as observed by [5]. Better terrain-flattening results using higher DEM oversampling multiples are achieved at the expense of longer

processing times and more computational resources. The area projection approach [5] has been shown to have better properties than the bilinear weighting approach of [4] in terms of computational resources and efficiency. In the future, we also plan to study the area projection implementation in ISCE3 [16] in the presented framework.

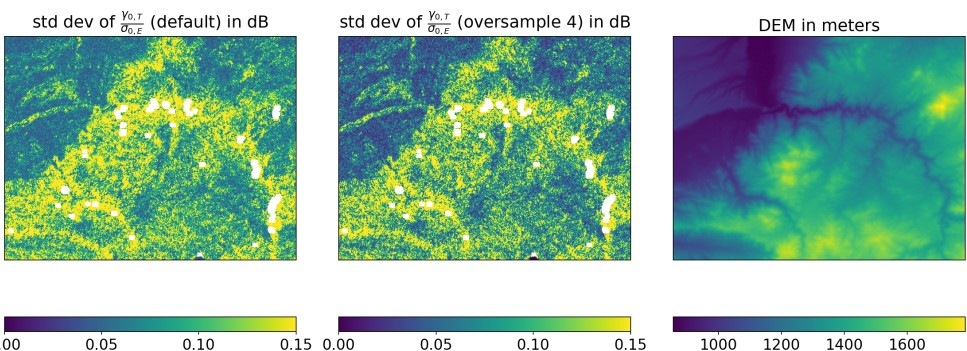

**Figure 9.** Standard deviation of the ratio of $\gamma_{0,T}/\sigma_{0,E}$ with masked out regions in white for default oversampling factor (**Left**) and an oversampling multiple of 4 (**Middle**). Corresponding DEM over a rugged 15 km × 15 km area near Big Bear, California (**Right**). An oversampling multiple of 4 reduces the observed processing error but does not eliminate the correlation with topography.

### 5.3. Global Terrain-Flattening Product

In [10], we described our pipeline for generating the near-real-time SAR backscatter $\sigma_{0,E}$ GTC product from Sentinel-1 data. Following the approach described above and in Section 3, one can also generate a global terrain-flattening factor product at a granularity of a single Sentinel-1 burst with the following layers using the Sentinel-1 toolbox:

- Static factor to transform $\sigma_{0,E}$ to $\gamma_{0,T}$ in decibel space.
- Shadow–layover mask

Such a global terrain-flattening factor product can be used to transform GTC products to RTC products on the fly using simple band-math in standard geospatial data frameworks. Additional useful layers such as local incidence angle, projection angle, line-of-sight angles, and nominal incidence angles can also be included for use with other workflows on the fly. For consistency, the computation of these layers should be done with the same processing engine as was used to geocode or terrain-correct the imagery. These global layers are significantly smaller in volume than the imagery archives, and can easily be distributed openly for efficient use in cloud-enabled geospatial frameworks.

## 6. Discussion

### 6.1. Applicability of Terrain Flattening

Although the terrain flattening of SAR imagery is widely considered to be a prerequisite for several applications, several applications do not necessarily require it. We discuss some cases below:

- Equations (1) and (3) clearly show that terrain flattening can be considered to be a correction of a pixel-by-pixel bias term. Consequently, if the analysis of individual SAR backscatter products can be reformulated as a ratio of polarization channels, e.g., radar vegetation indices, then the terrain-flattening effects are canceled out. Such analysis can be directly performed on GTC products.
- Section 3.1 shows that the pixel-by-pixel bias is consistent for narrow orbital tube missions. Consequently, if multi-temporal backscatter analysis can be reformulated to work with relative changes regarding a reference epoch or a temporal average, terrain-flattening effects are canceled out. This is similar to using a reference epoch in InSAR time-series analysis.

- Terrain-flattened products from different imaging geometries, e.g., ascending vs. descending passes, are not necessarily comparable over heterogeneous terrain such as urban areas, where the scattering mechanism is not necessarily distributed in nature. Comparing GTC products acquired from similar imaging geometries would allow for more sensitive change detection.
- Multi-temporal and multi-modal change-detection frameworks are becoming increasingly popular for wide-area monitoring and change-detection applications, e.g., [29,34]. These frameworks are designed to analyze time-series from multiple types of sensors and combine change detections. The sensitivity of change detection from SAR data can be improved just by considering different imaging geometries as different sensors in such frameworks.

### 6.2. Efficient Processing

As of 1 January 2023, Sentinel-1 has imaged over 50 million bursts in Interferometric Wide Swath (IW) mode. However, the number of unique burst footprints that have been imaged in the same time frame is only about 341,000 out of the possible 1,127,661 [12]. We can reduce the computation required for terrain flattening by a factor of ~150, using static flattening factors. A new flattening factor for a burst footprint only needs to be computed when a previously unimaged burst is imaged by Sentinel-1. The use of static flattening factors allows terrain flattening to be implemented on the fly using simple band-math within standard geospatial frameworks. The computational resources that are required to generate a GTC product are also significantly less than those needed to generate an RTC product, thus reducing the costs and resources needed to keep up with the live stream of Sentinel-1 data.

We also observe that using static flattening factors is the same as creating a stack of coregistered data with the Sentinel-1 toolbox before terrain flattening. This method uses the slant-range geometry of the reference image of the stack to terrain-flatten all the images after aligning them. The stack coregistration process is performed in slant-range geometry in the traditional implementations, and is computationally more expensive than the geocoding approach, which results in a coregistered stack in map coordinates, as described in [10]. Stacking is the approach recommended by Sentinel-1 toolbox developers to minimize geolocation inconsistencies, and our proposed approach produces similar results while requiring far fewer resources and computations.

One of the big advantages of the proposed approach of using static flattening factors is that these factors can be easily re-estimated efficiently and cost-effectively at a future date, with the same underlying SAR metadata and DEM if a better method were to be developed, without having to reprocess all the backscatter imagery data globally. Decoupling terrain flattening from terrain correction or geocoding allows us to support a wider range of applications with SAR data more efficiently and cost-effectively.

### 6.3. Validation of Terrain-Flattening Processors

Software implementation differences in terrain-flattening processors, particularly over sloped terrain, have also been observed in previous comparative studies [1,35]. Similar comparative and validation studies are also needed for shadow–layover mask generation approaches, as masking is an important aspect of robust multi-temporal change-detection applications with SAR data in general. The Gamma flattening approach [4] involves many more interpolation operations, both geometrically and over imagery, than our proposed approach (Section 2), which only involves geometric interpolation. Identifying the exact source of possible discrepancies observed in Section 5 will require more detailed comparative studies and the development of more comprehensive synthetic tests and metrics. These topics are beyond the scope of this manuscript. We again emphasize that metrics similar to the ones presented in Section 5 will be useful for the end-user community to understand the limitations of the underlying terrain-flattening implementation, open or commercial, that might be used to generate large datasets. Without synthetic test metrics, it will be hard

to compare results and identify discrepancies between implementations in comparative studies, as observed in [1,35].

### 6.4. Analysis-Ready Data Interoperability

The Committee of Earth-Observation Satellites (CEOS) is currently working on the standardization of five different families of Analysis-Ready Datasets (ARD) derived from SAR imagery (https://ceos.org/ard/, accessed on 1 January 2023).

- Normalized Radar Backscatter (NRB)
- Interferometric Radar (InSAR)
- Geocoded Single-Look Complex (GSLC)
- Polarimetric Radar (POL)
- Ocean Radar Backscatter (ORB)

These different families of products are often generated by independent processing chains and different Level-1 data as sources, e.g., NRB from Ground-Range-Detected (GRD) products and InSAR from SLC products. A lack of synchronization or cross-validation of these independent processing paths can lead to issues in the interoperability of these datasets not limited to geolocation offsets. Our proposed approach of using static terrain-flattening factors within the framework implemented in [10], ensures that all these families of products listed above are efficiently and consistently processed. In addition to these, our GTC $\sigma_{0,E}$ products [10] can directly support sea ice, soil moisture, and oceanography science users who do not typically work with terrain-flattened SAR data. Our proposed approach of using static factors works equally well for transforming GTC products to $\sigma_{0,T}$ and projected local incidence angle (PLIA) normalized products, e.g., [19,36].

### 6.5. Common Framework with InSAR

ALOS-1 is considered an extreme case in terms of baseline variation for modern InSAR-capable SAR sensors. Since most modern-day SAR sensors, e.g., TerraSAR-X, COSMO-SkyMed, ERS, EnviSAT etc., all exhibit much smaller baseline variation than ALOS-1, we can argue that the presented framework for flattening GTC products to RTC products using static flattening factors will work with all of these InSAR-capable sensors. We have also observed that the use of baseline information is a standard feature in InSAR time-series analysis, e.g., [37], and the presented approach of transforming GTC products to RTC products can be easily incorporated into the same InSAR time-series analysis tools. We can further simplify the transformation process and reduce it to band-math operations in standard geospatial frameworks once baselines and incidence angles are represented as coarse three-dimensional cubes [38].

## 7. Conclusions

In this manuscript, we have described the design principles and implementation details of a method to transform GTC SAR products into RTC SAR products. We have described a testing framework that relies on actual SAR metadata but uses synthetic imagery to validate terrain-flattening implementations and argue that similar frameworks should be more widely used to quantify and characterize processing errors in terrain-flattened datasets. Using this testing framework, we have demonstrated that static flattening factors, such as one-per-imaging geometry, are more than sufficient for efficiently flattening SAR imagery for terrain effects from missions characterized by a narrow orbit tube such as Sentinel-1 on the fly. We have argued that using static factors can reduce the computational resources needed to generate global-scale Sentinel-1 terrain-flattened products by a factor of ~150 compared to traditional image-by-image Gamma flattening approaches. The presented approach is efficient, cost-effective, and highly scalable; and is suited for handling, in near-real time, large volumes of SAR data that are expected to be acquired by missions such as Sentinel-1, NISAR, ALOS, and other InSAR-capable missions in the near future.

**Author Contributions:** Conceptualization, S.A.A., P.S.A., M.S.W. and M.T.C.; methodology, P.S.A. and M.S.W.; investigation, P.S.A.; data curation, P.S.A. and M.S.W.; software, P.S.A., M.S.W. and M.T.C.; validation, P.S.A., M.S.W. and S.A.A.; writing—original draft preparation, P.S.A., M.T.C. and M.S.W.; writing—review and editing, P.S.A., M.T.C., M.S.W. and S.A.A.; visualization, P.S.A. and M.T.C. All authors have read and agreed to the published version of the manuscript.

**Funding:** This research received no external funding.

**Data Availability Statement:** The single-look complex (SLC) Sentinel-1 and ALOS-1 imagery and associated metadata are available at the Alaska Satellite Facility's Vertex Portal here: https://search.asf.alaska.edu/#/, accessed on 1 January 2023.

**Acknowledgments:** The authors would like to thank members of Descartes Labs Applied Science group for providing useful feedback during pipeline development. We thank John Truckenbrodt and Marcus Engdahl for providing useful feedback on our Sentinel-1 toolbox related experiments.

**Conflicts of Interest:** The authors assert that they have no conflict of interest.

## Appendix A

Let $\vec{R}_{sat}$ represent the zero-doppler imaging position [9] of the SAR platform corresponding to target $\vec{T}$ in a given SAR acquisition of interest. Let $\vec{R}_{sat}^{ref}$ represent the zero-doppler imaging position of the SAR platform corresponding to $\vec{T}$ in the reference acquisition. Then, the interferometric baseline vector $\vec{B}$ in 3-D cartesian space corresponding to target $\vec{T}$ regarding to the reference imaging geometry is given by

$$
\begin{aligned}
\vec{B} &= \vec{R}_{sat} - \vec{R}_{sat}^{ref} \\
&= B_{\parallel}^{ref} \cdot \hat{l}_{ref} + B_v^{ref} \cdot \hat{v}_{ref} + B_{\perp}^{ref} \cdot \hat{s}_{ref}
\end{aligned}
\tag{A1}
$$

where $\hat{l}_{ref}$ is the unit vector along the line-of-sight from $\vec{T}$ to $\vec{R}_{sat}^{ref}$, $\hat{v}_{ref}$ is the unit vector along the velocity vector at $\vec{R}_{sat}^{ref}$ and $\hat{s}_{ref}$ is the unit vector along the normal to the slant-range plane. We refer readers to Figure 1 to understand the relative orientation of the normal to the slant-range plane regarding to the velocity and line-of-sight vectors. Please note that all these unit vectors are derived using the imaging geometry of the reference acquisition and are orthogonal to each other by design in a zero-doppler system. $B_{\parallel}$, $B_v$, and $B_{\perp}$ are referred to as parallel, along-track, and perpendicular components of the interferometric baseline, respectively. It is also important to note that interferometric baselines in this framework only depend on the position of the target $\vec{T}$ and are not affected by terrain slope at the target location.

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
