# Peer review of "Radiometric Terrain Flattening of Geocoded Stacks of SAR Imagery"

_remotesensing, doi:10.3390/rs15071932_

Round 1

Reviewer 1 Report

The authors present a highly relevant and topical technical note on RTC data from SAR sensors onboard satellites with stable orbit geometry. The method is centred on normalisation via facet area estimation to obtain RTC images from Sentinel-1 IW observations, taking up findings i.a. of Small 2011, Shiroma 2022, Navacchi 2022.

Overall, the study is worth publishing, contributing valuable results to current research, and is presented in clear language. However, several issues should be resolved before publishing this manuscript.

Introduction: I'm missing a statement on what are the benefits to users of RTC SAR. Help the reader to understand why this is worth the effort, and what is the potential of RTC SAR.

L42: sigma_0_E: The subscript "E" is unexplained (not clear at this point)

L55: Be more specific: GTC means the radiometry refers to the Ellipsoid, RTC means the radiometry refers to the Terrain/DEM. 

L64: Highlight the needed stable/precise orbit geometry, instead of stating "modern" SAR missions.

L72: I suggest: "ESA's" Sentinel-1 toolbox.

Figure 1: In my opinion visualizing the slant range plane is not helpful. Wouldn't it be better to visualize the plane perpendicular to the looking direction/line of sight, which is then used for normalising gamma nought? 

Figure 1: The bended line left of "E_01" looks odd to me.

L101: Do mean area calculation in 3D-space? If yes, I agree, and I suggest to state this there.

L107: typo: The "c"?

L116: This is not true, only for active shadow pixels. Pixels in passive shadow can have arbitrary incidence angles. If you meant only active shadow pixels then state this explicitly. 

L136: Yes, but the sampling of the DEM also plays a crucial role here. Please specify earlier in detail what map/grid you have chosen, and its sampling. 

Section 3.1: I'm missing a map here, a lot actully. How does the area/DEM look lie, how does the "burst" overlay? The term "burst" is new to me - I assume you mean (congruent and repeating) S-1 granule extent? L145: Also missing: reference or sketch what you mean with perpendicular baseline is missing.

L151ff: Nice and well-thought experimental setup! You nicely explain that the height for T00 is fixed to zero etc... However, you don't explain why you have chosen the lateral shifts of T01 and T10 and what the implications are. 

L170ff: Yes, but I think you still introduce a radiometric bias because the area stays only static on the ground, not in the orbit geometry where the area/energy should be applied in a weighted manner to normalise the measured radar brightness (as properly done by Shiroma et al 2022).

Section 3.2: again, a map would help a lot. Why California again? At least for Sentinel-1, Europe/Alps/Scandinavia would be an ideal testbed, also with an interesting overpass coverage: with areas overpassed by either 2/3/4 orbits next to each other.

L197: Why mention change detection applications? I would rather say its important to set this number in relation to the relative radiometric accuracy of the sensor. Please add some references here.

Section 3.3: This generalised formulation for non-narrow-orbit-tube SARs is isolated within this paper, and might be skipped in this publication. Also, an illustration showing the "B^ref" baselines is missing.

L277ff: Indeed. But perhaps more suitable in conclusions?

L309ff: How many scenes? A map would be helpful showing why you have chosen 150m-200m. Covers it every scenario on Earth? I guess significant topographic effects scale with the height variations of the terrain, e.g. in the Himalayan regions the impact of orbital deviations might be different than somewhere else.

L336: "8000": whis this magic number? please justify.

L338ff: Give a reference to the standard workflows.

L341: Specify "a UTM coordinate system". Did you use the EPSG-defined projections? For multiple UTM zones?

L373: Why? What's the impact?

Figure 6: Link the histograms clearly with the corresponding maps.

Figure 7: left: The mask looks very patchy. I think this might come from that SNAP only uses active shadow pixels too? center: Now a hint :-) ... I think this might likely come from the issue we have observed with SNAP: float32 instead of float64 for lat lon coordinates (https://forum.step.esa.int/t/geographic-coordinate-sampling/30591) 

L433ff: consider to move this to discussions/conclusions.

Section 6.1: Odd opener - L444 Does not support the motivation of your study! However: well argued within the points you mentioned!

L471: reformulate "order of 340 thousand"

Author Response

We would like to thank the reviewer for reading our original submission and providing valuable feedback. The comments and observations regarding the DEM oversampling strategy, in our opinion, has significantly help us improve the manuscript. Due to the length of the response, we have uploaded the original comments (in italics) and our response to the comments as an attachment.

Reviewer 2 Report

The paper is clear and well-written although I am missing in some of the parts to be more didactic for those readers who were profane with the basic concepts.

In the Introduction, the authors should add a couple of sentences that shortly describe what is a radiometrically-flattened SAR image and why it is necessary to have it.

Similarly, in Section 2 it would be nice to have a short description in simple words of the rationale behind the Gamma Flattening formulation.

When the DEM (and its facets) is projected to slant-range, are the authors using the orbits of the acquisition or jut the local incidence angles? The latter allows a simpler explanation but the former can be more precise with large scenes, like Sentinel-1 ones. Linked with the latter option, how the relative orientation of the DEM and the projection of the satellite orbit over ground (or in other words the heading of the image) is considered?

In my opinion, the paper can be published after the inclusion of the previous comments.

Author Response

We would like to thank the reviewer for reading our original submission and providing valuable feedback. Please find our responses to your comments below. The reviewer's original comments are shown in italics.

The paper is clear and well-written although I am missing in some of the parts to be more didactic for those readers who were profane with the basic concepts.

  • Thank you for the suggestion. We submitted  this manuscript as a “Technical Note” rather than an “Article” to the journal. We will try to address this as best as we can without migrating away from the spirit of a Technical Note. We have added some descriptions for basic concepts like terrain flattening (Section 1 - Introduction) and  have also provided an simple explanation of the traditional Gamma Flattening algorithm (Section 2.2). Hopefully, these new additions will address the reviewer's concerns.

In the Introduction, the authors should add a couple of sentences that shortly describe what is a radiometrically-flattened SAR image and why it is necessary to have it.

  • Thank you for this suggestion. We have added the following statement in the introduction section:

“SAR systems collect imagery in side-looking geometries and the resulting radar backscatter is determined by both the imaging geometry and the material properties of the area being imaged. For example, hills facing the SAR system are typically brighter as they scatter energy back towards the sensor and hills facing away are darker as they scatter energy away from the sensor. Analytic applications are concerned with the material properties of the area being imaged, not the imaging geometry. Modeling and reducing the contribution of the imaging geometry in the observed backscatter signal is called terrain flattening. “

Similarly, in Section 2 it would be nice to have a short description in simple words of the rationale behind the Gamma Flattening formulation.

  • Thank you for this suggestion. We have rewritten Section 2.2 based on comments from multiple reviewers and the start of this section now reads:

“The traditional Gamma Flattening approach [4,5] (see Figures~1-2 of [5]) tracks the mapping between individual DEM facets and pixels in slant range radar geometry using large lookup tables to account for the many-to-one mapping of area in map coordinates to slant range geometry. The total area of all contributing facets is accumulated before imagery in $\beta_0$ is normalized to $\gamma_{0,T}$. This many-to-one mapping between map coordinates and slant range coordinates almost always exists for each DEM facet as these two coordinate systems are not aligned. Use of large lookup tables significantly increases memory requirements and hinders parallelization of the flattening approach by requiring multiple passes over the map and radar image grids. When we start with GTC products, the calibrated backscatter measurements have already been aligned in map coordinates which presents an opportunity to significantly simplify the bookkeeping.”

When the DEM (and its facets) is projected to slant-range, are the authors using the orbits of the acquisition or jut the local incidence angles? The latter allows a simpler explanation but the former can be more precise with large scenes, like Sentinel-1 ones. Linked with the latter option, how the relative orientation of the DEM and the projection of the satellite orbit over ground (or in other words the heading of the image) is considered?

  • We use full orbital geometry based solutions for computing the area projections to slant range plane as well as the plane normal to the line-of-sight vector. For a single DEM facet case, both these approaches are equivalent and this allows one to verify that the implementation is correct. Our full geometry computations are the same as “geo2rdr” in ISCE and “SAT_llt2rat” in GMTSAR. All solutions are computed in 3D geometry taking into account the platform velocity (heading). This is also evident in our reference to the dependence on along-track baseline B_v in Section 3.3.

In my opinion, the paper can be published after the inclusion of the previous comments.

Reviewer 3 Report

The article falls into the scope of Remote Sensing journal, and it focuses on radiometric terrain flattering of geocoded stacks of calibrated SAR data for terrain-related effects.

Abstract contains structural description of main results and its affect on various types of data.

Introduction also includes a Terminology section, which is very useful for understanding the terms used in this study. However, I did not find clearly defined objectives of this study. In a scientific work, it is not necessary to mention "Manuscript Organization" - it is already clear with its traditional rules and should be followed, as is usually recommended by the editorial system, to make the study easy to understand.

The chapters “Revisiting the Gamma Flattening formulation”, “Terrain flattening of geocoded stacks”, “Impact of layover” and “Experiments with Sentinel-1 toolbox” are given instead of Materials and Methods and Results chapters. The authors decided on a non-traditional organization of the article written as a technical book, which makes it very difficult for the reader to orient in the text. I would also recommend not to use the phrase "we did", but to use the passive voice "was done" – in the whole contribution. I therefore recommend the authors to rewrite the otherwise well-described methods and results into the traditional form of a scientific article, as is customary in all quality journals.

Discussion - the study would help if it were more compared to other solutions.

Conclusions - should be in line with aims clearly define at the end of Introduction chapter. Also, the main results should be here mentioned clearly and concisely.

Maybe I explained well why I recommend to publish it with major revision.

Author Response

We would like to thank the reviewer for reading our original submission and providing valuable feedback. Please find our responses to the reviewer’s comments below. The reviewer’s original comments are shown in italics.

The article falls into the scope of Remote Sensing journal, and it focuses on radiometric terrain flattering of geocoded stacks of calibrated SAR data for terrain-related effects.

Abstract contains structural description of main results and its affect on various types of data.

Introduction also includes a Terminology section, which is very useful for understanding the terms used in this study. However, I did not find clearly defined objectives of this study. In a scientific work, it is not necessary to mention "Manuscript Organization" - it is already clear with its traditional rules and should be followed, as is usually recommended by the editorial system, to make the study easy to understand.

The chapters “Revisiting the Gamma Flattening formulation”, “Terrain flattening of geocoded stacks”, “Impact of layover” and “Experiments with Sentinel-1 toolbox” are given instead of Materials and Methods and Results chapters. The authors decided on a non-traditional organization of the article written as a technical book, which makes it very difficult for the reader to orient in the text. I would also recommend not to use the phrase "we did", but to use the passive voice "was done" – in the whole contribution. I therefore recommend the authors to rewrite the otherwise well-described methods and results into the traditional form of a scientific article, as is customary in all quality journals.

  • Thank you for your comments. This manuscript was submitted as a “Technical Note” and not as a traditional "Article" to the journal, which might explain the reviewer’s observation of the text reading like a technical book. It is possible that  the reviewer is expecting a standard academic thesis like document  structure and hence finds our document organization unconventional. The main paper that we reference Small (2011) is also very similarly structured to our manuscript. Our previous technical note Agram et al (2022) in the same journal also followed a similar structure. We also note that other reviewers have not highlighted the section titles as an issue. Hence, we would prefer to keep our current document organization to minimize disruption. We hope that the reviewer is willing to overlook the non-traditional organization and assess this manuscript as a “Technical note”.  We have also removed the section on “Manuscript Organization” as we have added text to the introduction to address a later comment. We have also tried to address the reviewer’s concern related to passive voice as best as possible.

Discussion - the study would help if it were more compared to other solutions.

  • Thank you for this suggestion. We note that previous comparative works - Truckenbrodt et al (2019) and  Ticehurst (2019)) found differences between different implementations (GAMMA and Sentinel-1 toolbox) of the same formulation but could not determine which of the two or both  implementations had errors. This is one of the reasons behind  advocating for synthetic tests to validate terrain flattening processor implementations. In our opinion, comparing our implementation to other solutions would not be very productive before each of the implementations that we compare against have also been validated with synthetic tests.  Else, we would not be able to ascertain the source of discrepancies. This is the motivation for the section regarding Sentinel-1 toolbox in this manuscript. There are 3 implementations of terrain flattening methods, that we are aware of, to potentially compare against:
    1. Sentinel-1 toolbox: We designed our experiments with Sentinel-1 toolbox in Section 5 and observed discrepancies in the synthetic tests and have reported it in this manuscript.
    2. GAMMA: This is a commercial software that we don’t have access to and hence are not able to compare our results against it.
    3. ISCE3: Shiroma et al (2022) released their algorithm in the ISCE3 framework recently but support for Sentinel-1 is missing. This is expected to be included in the next few months along with improvements to the implementation and we plan to evaluate this implementation as stated clearly at the end of Section 5.2.

We have also added the following sentence to the end of Section 6.3:

“Without synthetic test metrics, it will be hard to compare results and identify discrepancies between implementations in comparative studies as was observed in [1,35].”

We have also added a sentence on the importance of synthetic tests and metrics in the conclusions section as well.

Conclusions - should be in line with aims clearly define at the end of Introduction chapter. Also, the main results should be here mentioned clearly and concisely.

  • Thank you for the comments. The main results can be summarized as:
    • Development of an efficient method to transform GTC products to RTC products
    • Development of a testing framework for evaluating the quality of terrain flattening processors that uses actual SAR metadata but synthetic (constant valued) imagery
    • Use of the testing framework to validate our method
    • Observation that our proposed method can reduce compute resource requirements by a factor of ~150 for global scale Sentinel-1 RTC processing
    • Observation that more synthetic tests and metrics are needed for validating terrain flattening processors

We have updated the conclusions section of the manuscript to read:

“In this manuscript, we have described the design principles and implementation details of a method to transform GTC SAR products to RTC SAR products.  We have described a testing framework that relies on actual SAR metadata but uses synthetic imagery, to validate terrain flattening implementations and argue that similar frameworks should be more widely used to quantify and characterize processing errors in terrain flattened datasets. Using this testing framework, we demonstrate that static flattening factors, one-per-imaging-geometry, is more than sufficient for efficiently flattening SAR imagery for terrain effects from missions characterized by a narrow orbit tube like Sentinel-1 on-the-fly. We argue that using static factors can reduce the compute resources needed to generate global scale Sentinel-1 terrain flattened products by a factor of ~150 compared to the traditional image-by-image gamma flattening approach. The presented approach is efficient, cost-effective, and highly scalable; and is suited for handling, in near-real time, large volumes of SAR data that are expected to be acquired by missions such as Sentinel-1, NISAR, ALOS, and other InSAR-capable missions in the near future. “

We have also updated the Introduction section and added the following sentence at the end of Section 1.1:

“This manuscript presents a method to efficiently transform a GTC stack to an RTC stack using static flattening factors for SAR missions with orbit characteristics designed to consistently support interferometric analysis like Sentinel-1, ALOS-2, etc. We also present a testing framework for validating terrain flattening processors using Sentinel-1 SAR metadata and apply it to ESA's Sentinel-1 toolbox. Finally, we discuss how our proposed method can significantly reduce compute resource requirement for generating global scale Sentinel-1 RTC products while still generating these products in an inter-operable manner with other SAR-based ARD datasets.”

Maybe I explained well why I recommend to publish it with major revision.

Reviewer 4 Report

In this paper, authors provide detailed implementations for radiometric corrections considering height variations; this is coincided with transform GTC products to RTC products.

Because accurate radiometric interpretation is important especially in civilian applications, such kind of implementation is mandatory; thus, authors' studies are very useful for SAR processor engineers.

Overall, organization is good, and easy to follow. However, main contributions of this paper is not clearly described, compared to the previous related researches. 

To improve quality of this paper, thus, it is recommend to clarify key contributions of this paper in Introduction Section.

Author Response

We would like to thank the reviewer for reading our original submission and providing valuable feedback. Please find our responses to the reviewer’s comments below. The reviewer’s original comments are shown in italics.

In this paper, authors provide detailed implementations for radiometric corrections considering height variations; this is coincided with transform GTC products to RTC products.

Because accurate radiometric interpretation is important especially in civilian applications, such kind of implementation is mandatory; thus, authors' studies are very useful for SAR processor engineers.

Overall, organization is good, and easy to follow. However, main contributions of this paper is not clearly described, compared to the previous related researches. 

Response:

Thank you for the comments. The main results and contributions can be summarized as:

  • Development of an efficient method to transform GTC products to RTC products
  • Development of a testing framework for evaluating the quality of terrain flattening processors that uses actual SAR metadata but synthetic (constant valued) imagery
  • Use of the testing framework to validate our proposed method
  • Observation that our proposed method can reduce compute resource requirements by a factor of ~150 for global scale Sentinel-1 RTC processing
  • Observation that more synthetic tests and metrics are needed for validating terrain flattening processors

We have updated the conclusions section of the manuscript to read:

“In this manuscript, we have described the design principles and implementation details of a method to transform GTC SAR products to RTC SAR products.  We have described a testing framework that relies on actual SAR metadata but uses synthetic imagery, to validate terrain flattening implementations and argue that similar frameworks should be more widely used to quantify and characterize processing errors in terrain flattened datasets. Using this testing framework, we demonstrate that static flattening factors, one-per-imaging-geometry, is more than sufficient for efficiently flattening SAR imagery for terrain effects from missions characterized by a narrow orbit tube like Sentinel-1 on-the-fly. We argue that using static factors can reduce the compute resources needed to generate global scale Sentinel-1 terrain flattened products by a factor of ~150 compared to the traditional image-by-image gamma flattening approach. The presented approach is efficient, cost-effective, and highly scalable; and is suited for handling, in near-real time, large volumes of SAR data that are expected to be acquired by missions such as Sentinel-1, NISAR, ALOS, and other InSAR-capable missions in the near future. “

We have also updated the Introduction section and added the following sentence at the end of Section 1.1:

“This manuscript presents a method to efficiently transform a GTC stack to an RTC stack using static flattening factors for SAR missions with orbit characteristics designed to consistently support interferometric analysis like Sentinel-1, ALOS-2, etc. We also present a testing framework for validating terrain flattening processors using Sentinel-1 SAR metadata and apply it to ESA's Sentinel-1 toolbox. Finally, we discuss how our proposed method can significantly reduce compute resource requirement for generating global scale Sentinel-1 RTC products while still generating these products in an inter-operable manner with other SAR-based ARD datasets.”

To improve quality of this paper, thus, it is recommend to clarify key contributions of this paper in Introduction Section.

Response:

Thank you for this suggestion. We have now updated the last sentences of Section 1.1 to read:

“This manuscript presents a method to efficiently transform a GTC stack to an RTC stack using static flattening factors for SAR missions with orbit characteristics designed to consistently support interferometric analysis like Sentinel-1, ALOS-2, etc. We also present a testing framework for validating terrain flattening processors using Sentinel-1 SAR metadata and apply it to ESA's Sentinel-1 toolbox. Finally, we discuss how our proposed method can significantly reduce compute resource requirement for generating global scale Sentinel-1 RTC products while still generating these products in an inter-operable manner with other SAR-based ARD datasets.”

Round 2

Reviewer 3 Report

I would recommend authors to present the manuscript in the past tense, e.g. instead of "we describe" - "It was described" as is usual in scientific work.

Author Response

Original reviewer report:

I would recommend authors to present the manuscript in the past tense, e.g. instead of "we describe" - "It was described" as is usual in scientific work.

Our response:

We respectfully disagree with the reviewer. A simple abstract search for statements similar to what we used in our submission, for example,  "We present a new method" or "We describe a new method" will display numerous prominent and widely cited manuscripts.

Nevertheless, we have made the changes requested as they do not impact the technical content of the manuscript and to speed up the review process. We have gone through the manuscript text again and updated the wording when relevant to address the reviewer's feedback, as best as possible.